# The Potential Impact of a Public Health Approach to Improving the Physical Health of People Living with Mental Illness

**DOI:** 10.3390/ijerph191811746

**Published:** 2022-09-17

**Authors:** Russell Roberts, Caroline Johnson, Malcolm Hopwood, Joseph Firth, Kate Jackson, Grant Sara, John Allan, Rosemary Calder, Sam Manger

**Affiliations:** 1Faculty of Business, Justice and Behavioural Science, Charles Sturt University, Bathurst, NSW 2795, Australia; 2Melbourne Medical School, University of Melbourne, Parkville, VIC 3010, Australia; 3Division of Psychology and Mental Health, University of Manchester, Manchester M13 9PL, UK; 4NSW Ministry of Health, St Leonards, NSW 2065, Australia; 5Sydney Medical School, Northern Clinical School, University of Sydney, Sydney, NSW 2001, Australia; 6Queensland Health, University of Queensland, Brisbane, QLD 4072, Australia; 7Mitchell Institute, Victoria University, Melbourne, VIC 3000, Australia; 8College of Dentistry and Medicine, James Cook University, Townsville, QLD 4811, Australia

**Keywords:** comorbidity, mental health, physical health, COVID-19, cancer, CVD, respiratory disease, vaccination, smoking

## Abstract

With already wide disparities in physical health and life expectancy, COVID-19 presents people with mental illness with additional threats to their health: decreased access to health services, increased social isolation, and increased socio-economic disadvantage. Each of these factors has exacerbated the risk of poor health and early death for people with mental illness post-COVID-19. Unless effective primary care and preventative health responses are implemented, the physical illness epidemic for this group will increase post the COVID-19 pandemic. This perspective paper briefly reviews the literature on the impact of COVID-19 on service access, social isolation, and social disadvantage and their combined impact on physical health, particularly cancer, respiratory diseases, heart disease, smoking, and infectious diseases. The much-overlooked role of poor physical health on suicidality is also discussed. The potential impact of public health interventions is modelled based on Australian incidence data and current research on the percentage of early deaths of people living with mental illnesses that are preventable. Building on the lessons arising from services’ response to COVID-19, such as the importance of ensuring access to preventive, screening, and primary care services, priority recommendations for consideration by public health practitioners and policymakers are presented.

## 1. Introduction

The physical health of people living with mental illness is one of the most significant problems in global public health. For example, a 2017 study showed that in one year, 10,359 Australians with mental illness died prematurely of the top 10 (excluding suicide) causes of death [1]. On a population basis, this equates to one death per million population per day. Comprising 12.8% of the population, people accessing mental health-related services constituted over half of all premature deaths due to physical health conditions [2].

COVID-19 can cause a dramatic increase in the avoidable deaths of people living with mental illness unless targeted preventative action is taken. People living with mental illness already suffer a significantly elevated risk of early death from chronic health conditions such as respiratory disease, cancer, diabetes, and ischemic heart disease (see Figure 1) [3]. All of these conditions are associated with higher mortality in COVID-19 cases [4].

This paper briefly overviews key research related to the health outcomes for people living with mental illness with a focus on CVD, cancer, vaccine-preventable conditions, and smoking. Combining this research with population incidence data, the potential impact of public health interventions is modelled. It then examines major primary health and mental health policy positions and guidelines developed in the context of COVID-19.

The paper aims to build on recent research findings, COVID-19-related public health lessons, and policy recommendations to present four ‘shovel-ready’ public health interventions to improve the physical health of people living with mental illness.

## 2. Increasing Disparities in Health Service Access

Eighty percent of adults with a diagnosed mental illness have at least one comorbid mortality-related physical health condition, and 55% have two or more [5]. On average, people with mental illness in Australia die 14 years earlier than the general population [6,7]. There is substantial variability in the definition of severe mental illness in research [1,8,9]. Despite this definitional variance, there is strong and consistent evidence that people living with psychotic disorders such as schizophrenia have elevated risk and also that the increased risk of poor health and premature death is seen across the whole spectrum of mental health diagnoses [10].

Multiple factors increase the risk of early death for people living with severe mental illness due to the impact of COVID-19. These include reduced access to healthcare, increased risk of delayed (or missed) diagnosis, lower rates of vaccination [11,12], medication side effects, high rates of smoking [13], and, crucially, the presence of multiple long-term chronic health conditions [3]. Older people with mental illness are particularly at risk, given the vulnerability related to aging and the high rates of physical health comorbidity in older people with mental illness [2].

Generally, COVID-19 has resulted in fewer GP and emergency department visits for health concerns [14]. Further, the pandemic has reduced the capacity of the medical workforce, with increased burnout rates and COVID-19-related furlough periods. A delay in the diagnosis and treatment of chronic conditions such as heart disease, cancer, cerebrovascular disease, and diabetes will deleteriously impact patients’ short and long-term health outcomes. Previously, people living with mental illness tended to not access treatment due to reasons such as perceived stigma and cost (such as out-of-pocket expenses) [15]. Now they may also fear infection through exposure to the virus and increased difficulty accessing services due to the impacts of COVID-19 on the health workforce.

### 2.1. Heart Disease

People living with mental illness have lower rates of heart disease screening [16]. COVID-19 has resulted in a significant decrease in cardiac screening overall [17]. Heart disease causes the premature death of over 1740 Australians with mental illness every year, and most of these cardiac deaths are preventable [18]. Put in perspective, the number of preventable deaths of people with mental illness due to heart disease is greater than the total number of Australian deaths due to road accidents [19]. A New Zealand cohort study found that “risk factor patterns including metabolic disturbances may only be part of the reason for the high rates of CVD among people with SMI” [18] (p. 9). Diagnostic overshadowing of physical health conditions by mental health problems and differences in preventive care [18], screening [16], treatment [20], and procedural interventions [20,21] may contribute to this disparity, and each of these is less accessible in the context of the COVID-19 pandemic.

### 2.2. Cancer

People with mental illness have higher cancer mortality rates, even though their cancer incidence is the same or lower than in the general population [22]. In Australia, over 6000 people with mental illness die prematurely of cancer each year (see Table 1). This is more than the combined deaths due to suicide and road accidents [19]. There are already disparities in cancer screening for people living with mental illness [23,24]. The onset of COVID-19 has been associated with fewer GP visits, a decrease in cancer screening [25], and cessations of breast screening services [26]. (The greatest increase in relative risk of early death for women living with mental illness is breast cancer). Delayed detection and diagnosis reduce the chances of survival. COVID-19 can potentially increase both the rate and the proportion of cancer-caused deaths of people with mental illness [26]. Those with lung cancer have increased respiratory health risks associated with COVID-19, causing double jeopardy for these people.

### 2.3. Infectious Diseases and Vaccine Preventable Conditions

The existing data indicate that people living with mental illness have much lower hepatitis and flu vaccination rates than the general population [27]. The low vaccination rates for people with mental illness are unmistakably evident in the five times higher rate of vaccine-preventable hospitalization and seven times higher rate of vaccine-preventable hospital bed days [12]. People with mental illness are more susceptible to infections during epidemics. This is probably due to higher rates of medical co-morbidities, but it may also be related to confined living arrangements, lower levels of personal protection behaviors, and lower awareness of risk [28]. It can also be related to the increased difficulties in accessing timely medical services. In addition, when and if services are accessed, treatment is more challenging due to increased complexity related to the co-existing mental health condition.

One striking example during an infectious disease epidemic in Africa found that 49% of people living with severe mental illness died from an infectious disease [29]. Immediate public health action to provide people living with mental illness screening for current vaccination status, (including COVID-19 and influenza) would lessen virus prevalence in the community, reduce the burden on public hospitals, save money and save lives.

### 2.4. Smoking

The decrease in the rate of smoking in developed countries has been a public health success story, with the last 20 years seeing a 60% reduction in the percentage of the population who are daily smokers [30]. However, for people living with mental illness, smoking rates have remained high [13]. While international rates for smoking mostly fall in the range of 9–12% [30], smoking rates among people with mental illness are twice [31] to three times [30] that of those with no mental health disorder. Pre-existing respiratory disease and smoking are risk factors for more severe outcomes associated with influenza and COVID-19 [4]. Smokers who contract COVID-19 are 1.4 times more likely to have severe symptoms of COVID-19 and approximately 2.4 times more likely to be admitted to an ICU, need mechanical ventilation, or die than non-smokers [32].

## 3. Social Health, Physical Health and Mental Health

### 3.1. Social Isolation, Mental Illness and Early Mortality

People living with severe mental illness already suffer higher rates of social exclusion and social isolation [33]. The impact of social isolation on mental health is well documented [34]. However, social isolation is also a risk factor for all-cause (physical health) mortality [35], especially in the elderly [36]. Social isolation is related to increased cardiovascular diseases, cancer, and lower respiratory disease. These conditions are also the major causes of the early death of people living with mental illness [2]. Unless assertive action is taken, COVID-19-related social isolation will further amplify this already elevated risk of poor health and early death of people living with mental illness.

### 3.2. Increased Social Disadvantage

Many underlying causes of the reduced life expectancy for people living with mental illness are related to low socio-economic status (see Figure 2). As social inequality increases in a post-COVID-19 economy [37], health inequities will increase for people with mental illnesses. With COVID-19, the health disparities within the overall population will increase in relation to those with good cash reserves versus those without, those who can work from home versus those who cannot, those with work versus those who are unemployed, and those in commodious homes versus those living in crowded conditions [38]. These disparities are already present disproportionately in people with serious mental illness, and the data suggest this has increased due to COVID-19 [37].

Social disadvantage strongly affects health outcomes for the general population [37]. For people living with mental illness, this effect is amplified [39]. While people living with mental illness have 2.4 times increased risk of early death, for those with low SES, this increases to three times, and for those not in full-time employment, it is five times the risk (see Figure 2). While these data no doubt reflect both the causes and consequences of poor mental health, they nonetheless underscore the importance of good physical health care and the relationship between health and meaningful social and economic participation. In a post-COVID-19 economy, if these factors are not addressed proactively, the number of early deaths of people living with mental illness could increase significantly.

### 3.3. Physical Health, Mental Illness and Suicidality

The impact of COVID-19 has increased the prevalence of physical health problems. Poor physical health is associated with almost half of all completed suicides (see Figure 3) [40,41]. Despite extensive efforts during the last 2 decades, suicide has proven to be an extremely challenging problem to address effectively [42]. Improving the physical health of people living with mental illness should be a key component of any suicide prevention strategy.

Poor physical health also has a deleterious effect on mental health generally. Nearly all chronic medical conditions are associated with an increased prevalence of high psychological distress [43], with a dose–response relationship between poor physical health and depression consistent across 11 countries [44].

There is justifiably considerable attention on suicide prevention in the COVID-19 and post-COVID-19 context, and this should be expanded to include the (ten times greater) risk of early death due to preventable physical health conditions in people living with mental illness [2]. Further, physical health should not be overlooked as a key component of any COVID-19 or post-COVID-19 suicide prevention strategy. Improving the physical health of people living with mental illness is a largely neglected strategy to improve mental health and reduce suicide risk, particularly in older people [45].

## 4. Estimating Public Health Intervention Impact

Estimating the efficacy of public health interventions is difficult. Public health interventions need to be applied via policies or programs over entire populations, and the evaluation of epidemiological outcomes is obtained from observational studies. Efficacy is reliant not only on the resources needed but also on the social and organizational capacity needed to realize the conditions of change [46]. For instance, even if vaccine efficacy is high, the unit of analysis is a whole population, not individuals [47]. Further, the causes of the poor health and premature death of people living with mental illness are multiple, dynamic, and interrelated and exist within a changeable socio-economic environment. As such, estimating the efficacy of public health and clinical interventions is a complex process. Bearing this in mind, some discussion of potential public health program impacts across disease and intervention types is warranted.

### 4.1. Cancer and Cardiovascular Disease

With respect to ischemic heart diseases, people with schizophrenia are 70% less likely to undergo procedures and 40% less likely to be hospitalized, and 2.4 times more likely to die prematurely [48]. As such, interventions to achieve equity of access to screening, medical intervention, and hospital procedures might be expected to yield good results. On the other hand, the prescription of statins for reducing cardiovascular mortality has relatively low efficacy [49]. For cancer, even though people living with mental illness have the same or lower cancer incidence [22], they have eight times the rate of early death due to cancer [1]. Attaining equity of cancer outcomes would reduce the annual number of cancer-caused preventable deaths by 192 per 100,000 general population. However, the question remains as to what suite of interventions would be needed to move the rate of premature deaths due to cancer, heart disease, or other mortality-related chronic diseases down to the population base rate, and what capacity would be required to effectively implement a public health campaign across an entire population?

### 4.2. Smoking Cessation and Vaccination

Intervention trials have reported that people living with mental illness have a successful quit rate (at six months postvention) of 7–34% [50]. However, smoking cessation does not guarantee protection from premature death, and the effectiveness of a population-based intervention is expected to be much lower than in controlled trials [47]. With respect to the benefits of vaccination, based on hospital data, an effective campaign could reduce potential vaccine-preventable hospitalizations by 12.1 per 100,000 population per year [12]. Yet, this relies on successful implementation across the entire populace.

Based on these data, the efficacy of interventions such as vaccinations, smoking cessation, health screening, and medical interventions for mortality-related chronic conditions could be expected to lie between 0.1% and 25%. The actual efficacy of public health interventions depends on the clinical efficacy of the intervention itself, program level implementation effectiveness, the physical health conditions targeted, and the interactions between social, clinical, comorbid, and system factors. Table 2 presents the estimates of public health intervention impact on premature death rates, based on estimates of the proportion of preventable premature deaths (between 38% [51] and 69% [52]).

Investment in improving the physical health of people living with mental illness would significantly reduce preventable premature deaths of people living with mental illness. As an example, a 1% efficacy rate across the 38% of preventable deaths would result in an annual reduction of 3.14 deaths per 100,000 general population. (It should be noted that the 38% estimate of preventable deaths is based on a total population and, as such represents a conservative figure.) Put in context, for the populations of the USA, Japan, France, and Australia, this would mean an annual reduction in premature deaths of people living with mental illness by 10,626, 3894, 2029, and 822, respectively. The public value of such an investment is compelling and roughly equates to halving the suicide rates in these countries.

In addition to reducing preventable deaths, initiatives to improve the physical health of people living with mental illness would profoundly impact personal, social, and economic domains. Simple actions can make a discernible difference. For instance, effective vaccination programs for people with mental illness would significantly improve health, save hospitalization costs, and reduce pressure on health staff still struggling to deal with the impacts of COVID-19.

## 5. Discussion

The Lancet Psychiatry Commission blueprint for protecting physical health in people with mental illness and the WHO multi-level framework to address excess mortality in persons with severe mental disorders both underscore the fact that the major causes of the premature death of people with mental illness are multiple, complex, dynamic, and interrelated [53,54]. The major causes of preventable deaths include high rates of smoking, lower rates of physical health screening and treatment, medication side effects (particularly some anti-psychotic medicines), poor diet, and low levels of physical activity. Most of these factors are modifiable with the care workforce and infrastructure already in place. However, due to insufficient action across the domains of public health, illness prevention, primary care, and mental health care, the physical health disparities for people living with mental illness continue to increase [55]. Failure to implement basic health interventions proven to address these factors and improve health will increase the disparities in physical health outcomes for people living with mental illness due to COVID-19.

### 5.1. Policy and Practice Opportunities for Change

The first action of the Australian COVID-19 Primary Care Response (PCR) is to protect vulnerable people from the effects of COVID-19. People with severe mental illness are already at high risk of poor health and early death, so they must be proactively considered as some of the most vulnerable in this context. The PCR also commits to ‘continue the provision of regular primary care services to the whole community for acute and chronic conditions, and mental health concerns’ [56]. The physical health of people living with mental illness is also a significant component of the mental health pandemic plans [57]. Plans identify ‘those with chronic mental health concerns as being vulnerable to the impacts of COVID-19 including unemployment, poor physical health, social isolation’ (p. 20) and highlight their need to have access to regular or emergency health care [57].

Prior to COVID-19, a policy thinktank approached the problem of the poor health and early death of people living with severe mental illness using a primary care and chronic disease systems approach. It brought together national primary care, chronic care, and mental health experts from across Australia to develop a series of actions across the micro, meso, and macro levels of the health system. Arising from this process, the *Being Equally Well Roadmap* [58] recommended nine priority actions to improve the physical health of people living with severe mental illness. The recommendations included actions to enhance care coordination, such as guidelines and care navigators, removing financial barriers to access care, quality improvement mechanisms, additional research funding, stigma and discrimination campaigns, and targeted education material for health professionals.

### 5.2. Implications

The following suggested initiatives build upon the *Being Equally Well*, mental health pandemic, and primary care plan recommendations to address the increased vulnerability and risk of premature death of people with severe mental illness in the current COVID-19 context. Other well-mapped reform opportunities will improve the physical health outcomes for people living with mental illness, such as work engagement and community participation and programs to reduce the systemic discrimination experienced by people with mental illness in their contact with health services. While vitally important, these are complex system-wide challenges and require considerable time to develop and implement.

The health services’ responses to COVID-19 present an opportunity to apply the learnings regarding the importance of access to prevention, screening, and primary care to reduce the health inequity and life expectancy gap for people living with mental illness. The following recommendations are ‘shovel-ready’ actions that are readily implementable using the existing mental health and primary care workforce.

People living with severe mental illness should be included as a high-risk group with increased health vulnerabilities requiring targeted screening and treatment for high-risk chronic health conditions, particularly:
○Cancer;○Cardiovascular disease (see for example, the New Zealand, Cardiovascular Disease Risk Assessment and Management for Primary Care guidelines which recommend screening for people with severe mental illness over the age of 25 years) [59];○Respiratory disease;○Diabetes.People living with severe mental illness should be provided with subsidized access to all major approved vaccines (in addition to COVID-19 vaccines)People living with mental illness should have subsidized access to smoking cessation support, including nicotine replacement therapyIncreased support for telehealth follow-up of people living with severe mental illness and chronic physical health conditions, especially for older persons with mental illness.

### 5.3. Conclusions and Future Research

The current public health settings and actions appear to be not significantly reducing the life expectancy gap for people living with mental illness. The poor health and early death of people living with mental illness are partially related to the general population risk factor patterns related to mortality-related chronic conditions. However, it is also related to the stigma, systemic discrimination, and inequitable service access factors, which operate more strongly for marginalized groups such as those with experience of mental illness. There is a pressing need to implement effective public health interventions to address the poor health of people living with mental illness. This paper presented four relatively straightforward public health actions which are readily implementable and encompass both the general and specific risk factors associated with the major causes of poor health for people living with mental illness.

There is a plethora of research reports on the poor health and early death of people living with mental illness. Increased research focus on the efficacy of early detection and illness prevention programs for people with mental illness is needed. Research should also investigate the practice and process of implementing primary care or public health programs. In particular, research should explore the experiences of consumers, carers, clinicians, managers, and service partners involved in intervention trials, including their perceptions of program acceptability, validity, and practicality. This knowledge is crucial for the design and implementation of sustainable and efficacious programs.

People with mental illness have the right to ‘receive the same standard of health care as someone without mental illness’ [60] (p. 15). Reforms during and after the COVID-19 pandemic can ensure we respect these rights, enhance health, and improve the life expectancy of people living with mental illness.

## Figures and Tables

**Figure 1 ijerph-19-11746-f001:**
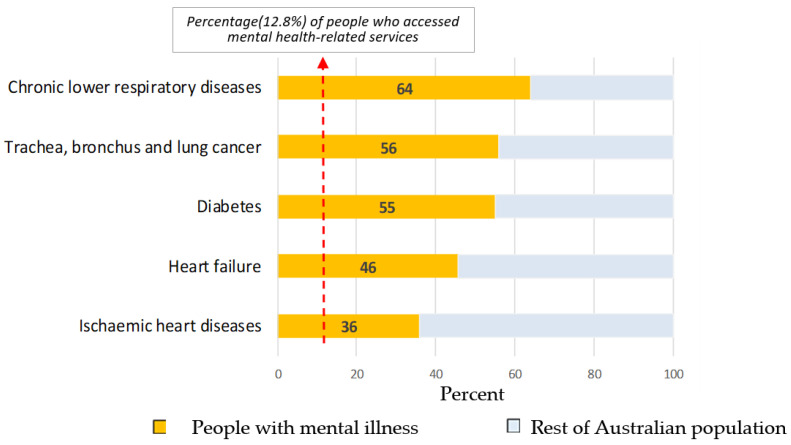
Proportion of early deaths of persons (age: 15–74 years) accessing mental health-related services by cause of death (adapted from ABS, 2017) [2].

**Figure 2 ijerph-19-11746-f002:**
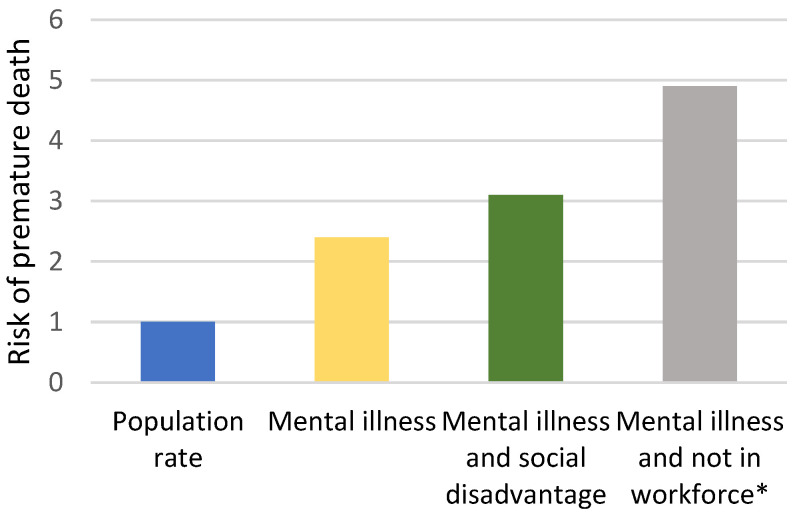
Relative risk of early death by population group (adapted from ABS, 2017 [2]). * Persons aged 15 to 64 years, compared to total Australian population 15 to 64 years.

**Figure 3 ijerph-19-11746-f003:**
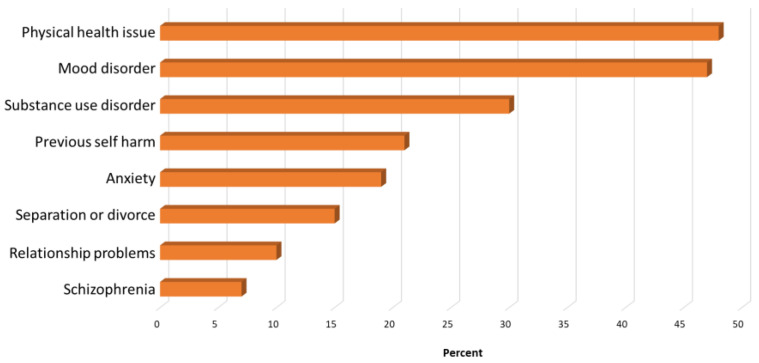
Percentage of suicide deaths associated with risk factors (adapted from ABS, 2018, 2019 [40,41]).

**Table 1 ijerph-19-11746-t001:** Annual number of cancer-caused deaths in the Australian population 2011/12—age 15–74 (prorated and adapted from ABS [2]).

Underlying Cause of Death	Mental Illness		Rest of Population		Total Population	
	No.	Row %	No.	Row %	No.	Row %
Trachea, bronchus, and lung cancer	2567	56	2031	44	4598	100
Colon, sigmoid, rectum and anus cancer	1064	51	1018	49	2082	100
Breast cancer	1012	56	785	44	1797	100
Blood and lymph cancer	895	51	876	49	1771	100
Prostate cancer	523	61	332	39	856	100
Total cancer caused deaths	6061	55	5042	45	11,103	100
					0	0
Total number accessing MBS/PBS	2.8 Mil	12	21.5 Mil	88	24.3 Mil	100

**Table 2 ijerph-19-11746-t002:** Estimates of annual preventable deaths of people accessing mental health-related services by preventability and treatment efficacy.

	Annual Deaths per 100 k			Annual Deaths per 100 k			Annual Deaths per 100 k		
		Preventable			Preventable			Preventable	
Intervention Efficacy	Total CVD, Cancer & Respiratory Disease	69%	38% *	Total Top 10 Causes (Excl. Suicide)	69%	38% *	Total All Causes	69%	38% *
	330	231.12	125.47	369	258.38	140.27	827	578.60	314.10
25%		57.78	31.37		64.60	35.07		144.65	78.52
10%		23.11	12.55		25.84	14.03		57.86	31.41
5%		11.56	6.27		12.92	7.01		28.93	15.70
1%		2.31	1.25		2.58	1.40		5.79	3.14
0.1%		0.23	0.13		0.26	0.14		0.58	0.31

* The 38% estimate is derived from the AIHW calculations for the entire Australian population [51], the 69% estimate is based on percentage of preventable cardiac deaths of people with serious mental illness [52].

## Data Availability

All data cited in figures and table is accessible through the supplied referenced and links.

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
