# Peer review of "The Potential Impact of a Public Health Approach to Improving the Physical Health of People Living with Mental Illness"

_ijerph, 2022, doi:10.3390/ijerph191811746_

Round 1

Reviewer 1 Report

The Authors present an interesting paper about the relationship between physical health. mental illness and COVID-19.

Line 22 should read: "increase in the post COVID-19" or increase after the  COVID-19"

Line 27 "based on" instead of "based"

Line 93-97 where in the referenced articles is the support for the Authors thesis that "the majority of these cardiac deaths are preventable" The reference states that the current CVD risk prediction equations are underestimating the actual incidence of CVD events in the mental illness population. Reference [20] says directly "Excess mortality is not substantially explained by treatment inequality." so it goes against the sentence in which the reference is used.

Author Response

Dear reviewer.

Thank you for your comments.

A response to each of the points raised is provided in the table below.

Reviewer 1

Line 22 should read: "increase in the post COVID-19" or increase after the  COVID-19"

This sentence has been revised to read:

Each of these factors have exacerbated the risk of poor health and early death for people with mental illness post COVID-19.

Line 27 "based on" instead of "based"

Revised and corrected

Line 93-97 where in the referenced articles is the support for the Authors thesis that "the majority of these cardiac deaths are preventable" The reference states that the current CVD risk prediction equations are underestimating the actual incidence of CVD events in the mental illness population. Reference [20] says directly "Excess mortality is not substantially explained by treatment inequality." so it goes against the sentence in which the reference is used

Thank you for this correction – this section has been revised to more accurately reflect the multiple factors and unexplained factors related to CVD disparities for people with severe mental illness, and the citations revised, and corrected to accurately support the assertions,  to read:

A New Zealand cohort study found that “risk factor patterns including metabolic disturbances may only be part of the reason for the high rates of CVD among people with SMI”(p. 9) [18]. Diagnostic overshadowing of physical health conditions by mental health problems, and differences in preventive care [18], screening [16], treatment [20] and procedural interventions [20, 21] may contribute to this disparity, and each of these is less accessible in the context of the COVID-19 pandemic.

Reviewer 2 Report

This is a well-prepared manuscript. This commentary paper provided interesting insight on COVID-19 and physical health. 

The Authors should consider the following changes to improve the scientific soundness of the manuscript. 

1. Please clearly define the study aim/aim of the commentary e.g. at the end of section 1. 

2. In the Introduction section the Author may add 2-3 sentences on how the commentary was prepared - how the study objective was identified and which sources were screened/used to prepare this commentary. The reviewer is aware that this is not a review paper, but some basic methodological notes will be appreciated. 

3. Please consider 2-3 sentences on conclusions/practical implications of this manuscript.

4. Moreover, 2-3 sentences on further research needs/research perspectives will be helpful.

Author Response

Dear reviewer,

Thank you for your review and revision requests.

A response to each of you request is listed in the table below - and the new added text can be viewed in the attached file.

thank you

1. Please clearly define the study aim/aim of the commentary e.g. at the end of section 1

Added   -- Lines 70-72

 (See attached document)

2. In the Introduction section the Author may add 2-3 sentences on how the commentary was prepared - how the study objective was identified and which sources were screened/used to prepare

Added  - Lines 65-69

(See attached document)

3. Please consider 2-3 sentences on conclusions/practical implications of this manuscript.

Added -  Lines 352-362

(See attached document)

4. Moreover, 2-3 sentences on further research needs/research perspectives will be helpful

Added – Lines 363-370

(See attached document)
